# Theoretical Challenges and Social Inequalities in Active Ageing

**DOI:** 10.3390/ijerph18179156

**Published:** 2021-08-30

**Authors:** Per H. Jensen, Jakob Skjøtt-Larsen

**Affiliations:** 1Department of Politics and Society, CARMA, Aalborg University, 9220 Aalborg, Denmark; 2Department of Sociology and Social Work, CASTOR, Aalborg University, 9220 Aalborg, Denmark; jsl@socsci.aau.dk

**Keywords:** active ageing, preconditions for active ageing, social life biography, health

## Abstract

Active ageing has been discussed in international political organisations and among researchers as a major means for combatting the challenges of demographic ageing. This study aims to make a critical-theoretical and empirical assessment of the active ageing concept, challenging the active ageing discourse from two different angles. First, an assessment of the theoretical framework of active ageing shows that the conceptual framework is undertheorised, lacks conceptual and analytical clarity, and fails to propose clear contributing factors and barriers. The second part presents an empirical analysis of the concept of active ageing guided by the following research question: is active ageing realistic—and for whom? Using Danish data subjected to multiple correspondence analysis, it is found that active ageing at the individual level is preconditioned by health, education, having good finances, etc. Furthermore, a Matthew effect of accumulated advantage is found; that is, older adults who are blessed in one sphere of life are also blessed in others, and such inequalities in old age are the outcomes of social life biographies (i.e., cumulative advantages/disadvantages over the life course). Thus, empirical findings indicate that active ageing may be an elusive goal for a large segment of older adults.

## 1. Introduction

Over several decades, the “activity perspective” [1] has gained footing in social gerontology and political communities. The “activity perspective” represents new standards of ageing “well” by rejecting that old age is inevitably associated with deficits, decline, and disengagement; rather, the “activity perspective” holds that inactivity in old age is usually involuntary. Two strands of research in social gerontology support the “activity perspective”: “successful ageing” and “active ageing.” These two perspectives have different roots and meanings [2,3,4,5]. The “successful ageing” concept, which is predominant in US ageing discourses, focuses on e.g., appropriate lifestyles enabling wellbeing and the participation of older adults [6]. In contrast, “active ageing”—the subject of this article—is primarily a concept [1,5] widely used in Europe over the last 15–20 years.

Still, there is no consensus on the actual meaning of “active ageing” [1]. It is a polysemic concept, subject to different theoretical approaches, analytical perspectives, and empirical observations. Some have used the WHO [7] definitions of active ageing as a test case and found that psychological variables are important contributors to the construct (e.g., [8]). Others have developed a model of active ageing that focuses on the individual level, analysing the impact of active ageing on life satisfaction and how coping styles may predict active ageing [9], while Barrio et al. [10] claim that the discourse of active ageing has paved the way towards active citizenship in later life. It has also been analysed how the social policies of active ageing are not in agreement with older adults’ wishes and aspirations [11]. Thus, a multiplicity of micro and macro level approaches offers different theoretical and analytical guidance as to how to study active ageing.

In this article, our point of departure is the blueprint of active ageing that has been developed in the active ageing index (AAI) project, defining active ageing as a “situation where people continue to participate in the formal labour market as well as engage in other unpaid productive activities (such as care provision to family members and volunteering) and live healthy, independent, and secure lives as they age” [12]. The purpose of the AAI project is to measure and promote the realisation of older adults’ active ageing potential, and the project is financed jointly by the European Commission (EC) and the United Nations Economic Commission for Europe (UNECE) [13,14]. The AAI project occupies a prominent position in discourses about active ageing. It is highly influential politically, and it has become a central reference point in academic reflections about active ageing. Thus, the AAI project is considered (by some) to be a practical “manifestation” of active ageing ideation [15].

As the AAI project occupies a central position in discussions about active ageing, the aim of this article is to critically to scrutinise the conceptual and empirical framework of the AAI project, meaning that we are challenging the AAI. Our critical analysis of the AAI project is divided into two parts:

First, we assess how the AAI project has conceptualised the construct of active ageing. Thus the aim of the first part of this article is to present the theoretical framework of the AAI-project, and then to develop a *theoretical* critique and show analytical shortcomings in the construct of the AAI-active ageing concept.

Second, the developers of the AAI-project have not yet addressed whether inactivity among older adults is actually involuntary [11,16], which means that social inequalities in active ageing prospects and the structuring mechanisms behind such inequalities have so far been under-researched. In effect, the second part of this article is an *empirical* analysis guided by the following research question: is AAI-active ageing realistic—and for whom?

When we test the premises embedded in the AAI-project empirically, we make use of the conceptual framework developed in the AAI-project as our point of departure. Hence, we reconstruct the AAI-project in that we make use of the same concepts (the four domains: (1) employment, (2) participation in society, (3) independent living, and (4) capability to age actively), variables, operationalisation, and indicators of active ageing as the AAI-project.

In analysing empirically whether AAI-active ageing is realistic, we draw primarily on two hypotheses: first, we hypothesise that active ageing is dependent on individual resources, given that numerous studies have shown, for instance, that older workers’ labour market activities (the employment domain) are conditioned by factors such as health, education, and age. Second, given that health (belonging to AAI-domain 4: capability to age actively) may precondition labour market participation (AAI-domain 1: employment), we furthermore hypothesise that there is a spillover between the different domains or that a Matthew effect of accumulated advantages exists, indicating that older adults who are blessed in one domain or sphere of life are also blessed in others—and vice versa. These hypotheses indicate that active ageing is only for the blessed and most privileged older adults.

Our analytical endeavour differs from the AAI-project. The AAI-project seeks to identity active ageing achievements and room for improvements, while we analyse active ageing inequalities. We therefore make use of a different metric than the AAI-project. We have thus chosen to apply the multiple correspondence analysis (MCA), as this metric is especially suited to identifying patterns of social inequalities, as well as the main factors structuring such inequalities.

Unfortunately, we are not able fully to replicate the variables of the AAI project. We draw on the Danish Senior Data Base (DSDB), comprising 9965 interviews with older adults 52–87 years of age. Regrettably, survey questionnaires used in the AAI and DSDB projects do not match one-to-one, meaning that the data used in this article are an approximation of the AAI data.

As indicated, the article begins with a critical assessment of the AAI-active ageing approach. Then we move on to the empirical analysis, and the article ends with a discussion and conclusion.

## 2. The AAI-Active Ageing Framework and Its Shortcomings

Like the notion of active ageing in general, the AAI-active ageing concept is founded on the idea that active ageing is beneficial for all older adults and the wider community alike [12,17]. The wider community benefits because active ageing reduces the financial burden of demographic ageing on society. Individuals benefit because participation in the formal labour market, as well as engagement in other unpaid productive activities (such as caring for family members and volunteering), is expected to empower older persons and allow them fully to realise their potential [12,18].

More specifically, the AAI project is designed to monitor developments and progress in the realisation of active ageing within the EU area. The project is based on the assumption that active ageing occurs within four domains in society: (1) employment, i.e., participation in the labour market; (2) participation in society, i.e., participating in voluntary activities, caring for children and grandchildren etc.; (3) independent living, i.e., doing physical exercise, access to health and dental care, independent living, etc., and (4) capability to age actively, i.e., share of healthy life expectancy at 55, mental well-being, use of ICT (internet use), etc. [5].

Within each domain, countries are ranked on a scale of 0–100, indicating country-specific achievements of active ageing. By crosscutting the four domains, an overall or aggregated index has been constructed that summarises all domains together. The overall ranking of countries in the ‘Active Ageing Index 2012’ showed that Sweden was the highest-ranking country (44.0 points) and Poland the lowest-ranking country (27.3 points) in Europe [14]. The developers of the AAI further argue that the distance from a maximum of 100 points indicates untapped potential [18] or room for improvements. This would mean that the index construction is based on the normative ideal that all countries score 100 in all four domains, which is unrealistic as Sweden currently reaches only 44 points.

This raises an index construction issue. Critics of the AAI have argued that social inequalities (e.g., gender, ethnicity, class, age) most likely condition opportunities for active ageing [13,14,15,17,19,20,21], and that inactivity may mirror the fact that some segments of older adults are unable to comply with the norms of active ageing due to structural, social, physical, or mental limitations. As such, the question becomes whether the AAI index measures (a) structural properties of different societies that may be hard to overcome as an act of will or (b) achievements of individuals at the micro level—or “room for improvement” due to involuntary inactivity—of the countries.

Another index construction issue is that on the micro level individuals cannot be fully active in all four domains due to the limitations of time (cf. [20]). Thus, to comply with the logic of the AAI project, older adults are expected to participate in the labour market; care for grandchildren and frail, elderly relatives; do volunteer work; engage in political activities and lifelong learning; perform physical exercise; etc. However, the AAI index disregards the limitations of cross-domain activities, indicating that “room for improvement” may also be limited.

Furthermore, a basic question is whether active ageing is actually in accordance with older adults’ wishes, dispositions, and aspirations; that is, are older adults really *willing* to become active according to the ideas of active ageing? This is an empirical question not addressed by the AAI project. Sufficient financial security and good mental and physical health are most likely central AAI-indicators that would be widely supported by older adults. However, in relation to the employment domain, for example, it is safe to say that older workers’ orientations towards the labour marked differ markedly across Europe. The ISSP 2015 Work Orientations module [22] reveals that the proportion of older adults 56–65 years of age that “strongly agree” or “agree” with the statement that they would “Enjoy a paid job even if I did not need the money” was e.g., 67 per cent in Denmark and 34 per cent in Finland. With the Finnish figures in the back of one’s mind, it becomes hard to claim that all older adults all over Europe will feel themselves ‘empowered’ or that they have been fully allowed to ‘realise their potential’ if they are forced to work longer as an outcome of active ageing policies, such as pension scheme changes [23]. Of course, the decision to raise state pension ages all over Europe is not necessarily an external force pressuring older workers to continue working. Nonetheless, as state pension ages are rising, most older workers have no choice but to prolong their working lives.

This indicates that some dimensions of active ageing are more beneficial for the welfare state and employers than for older adults, often conceptualised as a productive notion of ‘active ageing’ [18]. It is therefore no surprise that the AAI project is fostered by the political system and managed and financed jointly by the European Commission (EC) and the United Nations Economic Commission for Europe (UNECE) [24]. Accordingly, the primary aims of the AAI project become the identification of public policies and programmes that may enhance or function as a prerequisite for active ageing [12,25,26]. Such findings are transported back into the political system, wherein knowledge about factors enhancing active ageing is supposedly used to rethink or reformulate strategies promoting active ageing [5]. As such, social science researchers working with active ageing are predominantly communicating with the political system, epitomised as a wide range of “stakeholders, such as policy-makers, researchers, students and businesses” [13].

It is sometimes argued that active ageing, in contrast to the successful ageing concept, represents a “sociological turn,” meaning that active ageing is not preconditioned by (internal) psychological features (as the successful ageing concept prescribes) but rather by external opportunity structures [5]. However, the AAI perception of active ageing is marked by major theoretical and analytical shortcomings. The AAI project is undertheorised, and the ideas informing the AAI vision are rarely discussed or justified [2,16,20]. From a theoretical point of view, concepts and distinctions seem arbitrarily constructed. For instance, domains—a central dimension of the AAI-active ageing framework—originate from an EU political discourse about “societies for all ages”, wherein the Council of the European Union has reaffirmed in 2012 that active ageing must “be promoted in the three domains of employment, participation in society and independent living” [12]. This clearly indicates that the choice of domains is politically informed and defined.

Hence, the AAI conceptual framework does not clarify the constituents or determinants of active ageing, which—at the individual level—may be largely dependent on individual resources. Within the “employment” domain, for instance, numerous studies have shown how the timing of retirement is conditioned by factors such as health, education, and age [27], while studies within the participation-in-society domain have shown that engagement in volunteer work depends on factors such as the participants’ levels of education [28,29]. Even Barslund, von Werder and Zaidi [30] have, within the framework of the AAI project, shown that vertical or hierarchical inequalities in active ageing do exist.

Horizontally, very little is known about the relationships between the different domains; for example, how does the “employment” domain interact with the “participation in society” domain? Such interactions likely do exist, and Asghar Zaidi, who coordinated the AAI project, and colleagues have made a few empirical analyses mapping the spillover from one domain to another. For instance, they have found a strong correlation linking Domain 4 (capability to age actively) to Domain 2 (participation in society) and Domain 3 (independent living) [25]. This suggests that different activities and individual resources are interdependent [19], meaning that a high score in one domain is associated with high scores in other domains. However, the structuring mechanisms behind such spillover effects are unknown. At the individual level, such interdependence is most likely intermediated by factors such as, e.g., educational attainment, given that (1) the timing of retirement, as well as (2) the participation in volunteer work as already mentioned, seems to depend on level of education.

When measuring the degree of active ageing within the four domains, it is obviously of utmost importance that there is a minimum of logic in the choice of indicators. However, the choice of indicators is marked by a lack of transparency and by an element of arbitrariness [15]. The developers of the AAI have chosen indicators that are simple, understandable, and comparable [12]. Potentially relevant indicators that cannot be compared across countries based on existing statistics are excluded. The choice of indicators was also influenced by comments and feedback from the UNECE Expert Group, which functioned as an advisory board on the AAI project [24]. Minutes from these expert meetings are publicly accessible (e.g., [31]). In the minutes, the “opinions” of experts are reported. But the foundation of these opinions and how they have been translated into the indices are not explained.

In the analysis, different domains and indicators have been assigned different weights, meaning that the choice of weights in the measurement is crucial for the results [21]. However, the developers of the AAI have made clear that the weights are “admittedly arbitrary” and without any “theoretical guidance” [25]. Rather, the weights “reflect the political relevance as perceived by the Experts” [24]. Thus the AAI project results represent normative judgements based on expert opinions, leaving us with the old Baconian epistemic problem, meaning that domains, indicators, and weights would have undoubtedly looked rather different if the advisory board had been differently composed, e.g., composed of Scandinavian feminist scholars engaged in ageing research.

Another basic problem of the AAI project is that it neglects differences in value systems and cultural orientation across Europe, i.e., it adheres to a unidimensional cultural model that is insensitive to cultural diversity across European countries [20]. For instance, in the Participation in Society domain, the score is high if older people function as carers for others, typically their own frail parents (or spouses) and their grandchildren. This, however, is based on norms and values predominant in southern European—and, to some extent, central European countries—and within the framework of the male breadwinner model, whereas in northern European countries like Denmark with its dual-earner family structure, it is the obligation of the welfare state to care for frail older persons [32]. In effect, Denmark scores much lower on indicators such as ‘care for children, grandchildren’ compared to e.g., Italy, because the Danish and Italian welfare states are founded on different ideals, which means that the developers of the AAI have developed concepts, distinctions, and reflections on goals and means that may be disconnected from the social reality in most European countries. One may therefore ask: what is the basic notion of society from which the AAI project develops its ideals and distinctions?

That the AAI project is disconnected from any coherent theory of society or systematic information about older adults’ dispositions, aspirations, and characteristics has the effect that there is no compulsive logic in how concepts and distinctions are evolved. Conceptual combinations are arbitrary. The AAI conceptual framework is nonetheless organised according to a hidden “logic”; namely, the logic of the dichotomous organisation of concepts, constituting the world as either “in or out”, “before or after”, “good or bad”, etc. The AAI project is thus based on “a positive normative judgement meaning that the higher the value, the better the active ageing outcome” [12], which has profound effects on researchers’ possible insights and theoretical positions. As researchers, we focus on what is measurable and on registering a position on one side or the other of a boundary. In this endeavour, we observe according to an “objective” criterion (e.g., older or younger than age 60) or by disregarding an “objective” criterion (man or women). This dichotomous approach has obvious advantages; it permits distinction between political intention and scientific observation. Intention is a political domain, whereas the scientific task is to register and deliver knowledge about achievements in active ageing, irrespective of the fact that the researcher’s choice of concepts and approach is informed by politicians, i.e., is normatively and politically motivated.

## 3. Materials and Methods

### 3.1. Design and Procedure

We do not propose to address empirically all of the questions raised above. Rather, we address whether AAI-active ageing is realistic—and for whom. We set out by using indicators approximating the same indicators as the ones applied in the AAI model. We test whether active ageing opportunities are unequally distributed among older adults, and we test how activity in one domain is related to activity in other domains. In this endeavour, we make use of Danish data, i.e., data from a country with a highly developed social democratic welfare state. Denmark has one of the lowest Gini coefficients within the OECD area, and Denmark is among the highest-ranking countries in all four domains.

In order to study inequalities and their structuring mechanisms in relation to active ageing, this article applies multiple correspondence analysis (MCA) to cross-sectional data. A number of different methods are available for reducing complexity in data and for identifying dimensions across a number of single indicators in cross-sectional data (i.e., factor analysis, principle component analysis (PCA), and MCA). MCA is often described as a close analogue of PCA [33]. Like PCA, MCA identifies a number of principal axes, with the first principal axis providing the best one-dimensional fit of the data in the orthogonal least square sense. Adding the second axis creates the best-fit plane, and so forth [33]. Unlike PCA, MCA is specifically developed to handle categorised data, which makes it particular attractive for the analysis of survey data with categorised answer options.

### 3.2. Data

This article draws on data from the fourth round of the Danish Senior Data Base (DSDB) from 2012. The DSDB is a survey-based longitudinal study, which seeks to map the living conditions among the elderly of today and tomorrow. In 2012, 9965 interviews were conducted with people ranging in age from 52 to 87 [34].

The response rate in 2012 was 74%. The study is characterised to some extent by socio-economically imbalanced response rates. People with limited education, income, and wealth are overrepresented among the non-respondents. In terms of health, when using Charlson’s comorbidity index, the differences in the response rate are significant without being large. The differences are somewhat greater when considering only mental health. After controlling for demographics, finances, and age, seniors without a psychiatric diagnosis have a non-response probability of 25%, whereas those with a psychiatric diagnosis have a 38% non-response probability [35].

When using the DSDB data to analyse inequality in active ageing at the individual level, the AAI study and our analysis do not match one-to-one. First, indicators such as remaining life expectancy are impossible to measure at the individual level. We have therefore made use of alternative indicators. In the case of remaining life expectancy, for instance, we have used self-rated health with the question: “How do you think your health compares to others your age?” The rationale behind this choice is that an extensive literature has demonstrated that self-ratings of health do indeed predict mortality [36]. Similarly, it is impossible to measure relative median income at the individual level. Instead, we asked, “How do you think your finances are today?” Very good/good—OK—bad/very bad. Second, as in the AAI project, we have included indicators of gender and age. Overall, Table 1 shows how the original macro-level AAI indicators are linked to the DSDB individual-level indicators.

Table 1 shows that data selection is organised along the line of the four AAI domains, i.e., (1) employment, (2) participation in society, (3) independent, healthy, and secure living, and (4) capacity and enabling environment for active and healthy ageing. In addition, we have included data on (5) individual characteristics, such as gender and age.

Furthermore, the first column in Table 1 shows how domains have been operationalised in the AAI project, while the second column shows how we have operationalised the four domains based on DSDB data. The third column shows how data have been included in the statistical analysis. The indicators included directly in the statistical model are marked as “active,” whereas the indicators included in a secondary analysis are marked “supplementary.” In MCA, the variance of the model is calculated using the categories of the active indicators only. The number of categories for each of the active indicators is listed in the fourth column.

We have moreover decided not to make use of the AAI-specific weights, primarily because they have been chosen arbitrarily. Furthermore, the developers of the AAI-index have actually analysed the AAI data using two different weighting systems, and different weighting systems were not found to produce markedly different AAI scores [12].

### 3.3. Principles of Multiple Correspondence Analysis

The method chosen for analysis is multiple correspondence analysis (MCA). MCA is a descriptive statistical technique that is particularly well suited for identifying patterns in qualitatively coded data, such as nominally or ordinally coded survey data [37]. Correspondence analysis is a statistical technique based on multidimensional geometry. It allows for the identification of patterns and relationships in large datasets visualised in a coordinate system (e.g., [38]). MCA is based on a geometrical principal representing data as sets of points in a multidimensional Euclidian space [33]. Performed on survey data, MCA results in a cloud of individuals and a cloud of categories. Interpretation takes place using the categories that are visualised in two-dimensional coordinate systems. On each axis of a coordinate system, categories (answers to questions in the survey) that are frequently found to co-exist within the same response profiles (same individuals) will be situated close to each other, while categories that seldom occur within the same response profiles will be situated far from each other [33,37]. In other words, if respondents who score high on wellbeing are likely to also use the internet but not to visit their doctor, the categories representing the first two responses are likely to be positioned in proximity to each other with the category representing the latter response at a distance. In MCA, a distinction is drawn between “active” and “supplementary” elements. The structure appearing in the space is based solely on the active variables (e.g., use of ICT), and the distances between the categories (e.g., use often) in the model are calculated from them. Having interpreted the axes based on the distribution of the “active” categories of an axis, it becomes possible to insert additional “supplementary” variables. The categories of the “supplementary” variables are treated as categories with zero weight, meaning that they do not contribute to the distances calculated in the model [33]. This technique essentially makes it possible to include additional variables in the analysis and to inspect the distribution of their categories along the axes already established using the “active variables.” To determine if a supplementary variable is related to an axis, Le Roux and Rouanet [33] suggest that deviations between the categories of a variable on a given axis above 0.5 may be considered “notable,” while deviations > 1 are large [33].

## 4. Results of the Empirical Analysis of Active Ageing

The first step in the analysis is to select a relevant number of axes for interpretation. Each of the MCA axes sums up a decreasing part of the total variance of the model. In our model, there are marked reductions in the explained variance after the third axis, indicating that three axes can be interpreted. We present the first three axes summing to an accumulated modified rate of 68%. Axis 1 is related to all of the active-ageing domains, indicating a correlation between indicators from the different domains. The variables contributing the most are ICT (24%), independent living arrangements (16%), and taking care of older adults (13%). The variables contributing the most to Axis 2 are self-rated health (37%), mental wellbeing (24%), and access to health care (18%). Axis 3 primarily relates to physical exercise (26%) and self-rated health (23%) but also to voluntary activities (13%) and independent living arrangements (13%). Categories contributing above average (100/20 = 5%) are retained for the analysis of the main opposition of each of the first three axes. Categories contributing above average to the variance of Axes 1 and 2 are included in Table 2 and Table 3 as well as in Figure 1. Categories in brackets contribute less than average but are included as oppositional categories to aid the interpretation [33]. The Axis 3 categories will not be displayed, but a summary interpretation will be presented. The axes of MCA are orthogonal to each other [33,37], and the axes we interpret are uncorrelated. Axis 1 sums up the most important polarities in the data; Axis 2 the second-most important, etc. [37]. Hence, the plane constructed by crossing Axis 1 and Axis 2 (Figure 1) gives the best low-dimensional representation of the patterns in active ageing, given the indicators included in the model.

Interpretation of Axis 1 (λ1 = 0.175): The categories contributing above average to Axis 1 can be read from Figure 1 (horizontal line) and Table 2. To the right of Axis 1 are indicators of actively using ICT (use of ICT: use often), not living by oneself (independent living: don’t live alone), taking care of older adults (care for older: yes), doing voluntary activities (voluntary activities: yes), doing physical exercise (physical exercise: active), and having better self-rated health than most (self-rated health: better than most). On the left side are indicators of the opposite. We interpret this as an opposition between being both socially and physically active and well versus being inactive and less well. Turning to the supplementary elements projected in Figure 2, being well and active is positively associated with high levels of political participation (Figure 2c), high levels of education (Figure 2e), having been part of the salariat rather than the unskilled or skilled occupations in the labour market (Figure 2f), still being in the labour market (Figure 2b), making ends meet (Figure 2d), and reporting a good or very good financial situation (Figure 2g). The pattern along Axis 1 is not significantly related to age or gender.

Summary interpretation: We interpret this axis as a general axis summing up oppositions regarding *Health and active living*. It separates the socially and physically active and well from the socially and physically inactive and less well. It follows from the supplementary elements that being healthy and active in general is associated with economic and educational privileges, but it is not related to age or gender.

Interpretation of Axis 2 (λ1 = 0.133):

The categories making above-average contributions to Axis 2 can be read from Figure 1 (vertical line) and Table 3. On the lower side of Axis 2 are being mentally well (mental wellbeing: high) and in good health (self-rated health: better than most; access to health care: did not visit). Not having caring responsibilities for others only contributes slightly above average to this pattern. Along the top side in Figure 1 are opposing answers to the same questions. We interpret this pattern as an opposition between being in good mental and physical health versus being less well mentally and physically. Turning to the supplementary variables (Figure 2), being mentally and physically well is associated with having good finances (Figure 2g) and is (perhaps somewhat surprisingly) positively associated with age, indicating that older respondents are more likely to evaluate their mental and physical health as being relatively good (Figure 2h).

Summary interpretation: We interpret this axis as specifically related to *health* (and to some degree with not having caring responsibilities for elders). It separates the mentally and physically well from the mentally and physically less well, with the former being positively associated with income and age.

Interpretation of Axis 3 (λ1 = 0.126): Axis 3 is not illustrated. On the lower part of Axis 3, we find that indicators of being physically and socially active (physically active: active; voluntary activities: yes) corresponds to indicators of fair health (self-rated health: as good as most; access to health care: did not visit) and with living alone (independent living: live alone). The opposing answers to the same questions are on the top side of Axis 3. We interpret this pattern as an opposition between being physically and socially active versus being less active in these regards. Turning to the supplementary variables, we find that being active is positively associated with being politically active but not with any other enabling or individual characteristics.

Summary interpretation: We interpret this axis as a general axis summing up oppositions related to *physical and social activity*. It separates the physically and socially active living alone from the physically and socially inactive living with others. The axis does not appear to be influenced by individual characteristics (apart from also being active in politics). We consider this pattern to be a specific variant of the one identified in Axis 1. It shows opposition between the physically and socially active to the less active, but unlike Axis 1, being active is associated with living alone and with health being as good as others (rather than being better than others).

## 5. Discussion

The notion of active ageing has gained footing in discourses about how to meet the challenges from demographic ageing. The message is that “activity” is beneficial for society, as well as desirable for all older workers, and that structural or individuals’ constraints for the activity paradigm are limited. As such, active ageing is considered (more or less) to be a purely technical question (a technicality) that can be handled top-down through prudence, reason, insights, and declarations of intent.

There is no consensus about how active ageing can be framed theoretically or analysed empirically. However, as the AAI-project occupies a central position in the study of active ageing, we have critically scrutinised the AAI-project to come to grips with some of the basic problems embedded in the notion of active ageing and how it is studied. We have thus made a second order analysis of the AAI-project from two different angles: a theoretical and empirical.

A basic theoretical problem is that the concept of active ageing is nurtured by the political system and that the AAI-project is driven financially by the EU and the UN. This means that the AAI-project is both informed by the political system and designed to report back to the political system by identifying policies that enhance active and healthy ageing [12]. As such, the AAI-project is guided by political ambitions (AAI may be said to be a political benchmark tool) rather than standards of science (objectivity and transparency). Therefore, it may not come as a big surprise that the AAI-active ageing conceptual framework is undertheorised and lacks coherence and conceptual and analytical clarity.

Of course, researchers making use of the concept of active ageing, including the developers of the AAI-project, are all fully aware of this basic problem. The answer thus far has been that researchers who make use of the concepts have argued for improvements in the methodology, such as rethinking the weighting and showing respect for cultural diversity [21]. This also points to a limitation of our critique, namely that the critique is based on the implicit assumption that the AAI is a scientific—and less political—contribution. In any case, future research should focus on strengthening the idea of active ageing theoretically.

Empirically, the indicators used in the AAI-project supposedly measure achievements (or room for improvements) in active ageing, and the AAI encourages countries to develop policies that facilitate the participation of older adults. It remains unclear, however, which policies and programmes actually promote participation and engagement among older adults. For instance, the AAI-project does not give an account for which premeditated policies of the past (or type of society) have led to the fact that Sweden has an active ageing score of 44.0 points, while Poland only scores 27.3 points. Future research should therefore be more engaged in identifying policies that are conducive to active ageing and transferable from one country to another.

The active ageing discourse takes for granted that older adults are willing and able to become active. As a result, the active ageing approach tends to neglect frailty and limitations. As shown empirically in this article, inequality in ageing is conditioned by factors such as class and wealth, i.e., factors rooted in the social life biography. Future research should therefore focus more strongly on early-life experiences and cumulative disadvantage over the life course [39] in the study of older adults’ active ageing prospects.

These findings are based on an empirical study where we replicated the conceptual framework of the AAI-project in order to address the weaknesses of the AAI approach on its own terms or from an AAI-internal position. However, this also reveals a weakness in our empirical study. We have not been able to replicate the AAI approach in a 1:1 relation. We have used data that are not fully compatible with the AAI data. For instance, in the domain of independent living, one serious limitation is that the AAI-project analysis “Percentage of people aged 75 years and older who live in a single household alone or in a couple household”, whereas we based our variable on independent living on the following survey question: “Do you live alone? Live alone/don’t live alone”. Finally, in adopting an exploratory approach to a single nation study, we make no claims that the specific dimensions (axes) of active ageing identified will be identical in other contexts.

## 6. Conclusions

The present study has examined the AAI-project as a prominent and prototypical example of studies in active ageing, made relevant by political discussions about how to meet the challenge of demographic ageing. We have (a) assessed the theoretical framework of active ageing propagated by the AAI-project, and we have, (b) at the individual level, empirically studied the feasibility of active ageing by posing the following research questions: is AAI-active ageing realistic—and for whom? The contribution of this study is the following:

The AAI-project is a political—and less scientific—contribution to the study of active ageing. Thus, the AAI construct of active ageing is structured along lines that seem to be relevant for the politicians and less suited for scientific (sociological) observation. The theoretical and conceptual framework of the AAI-project thus suffers from a lack of coherence and theoretical underpinning.

The empirical part of the study proves that active ageing is idealistic and unrealistic and ignores the life situation of large segments of older adults in that active ageing opportunities are conditioned on e.g., one’s health and position in the social structure. In accordance with findings by Westerlund et al. [40], we have also found that retirement (inactivity in the employment domain) has a positive effect on older adults’ health and life situation. Furthermore, using the AAI domain approach it has been shown that a clear Matthew effect of accumulated advantage exists: Those scoring high on one indicator/domain also score high on other indicators/domains. These empirical findings clearly are a challenge to most of the propositions embedded in the “activity” perspective.

## Figures and Tables

**Figure 1 ijerph-18-09156-f001:**
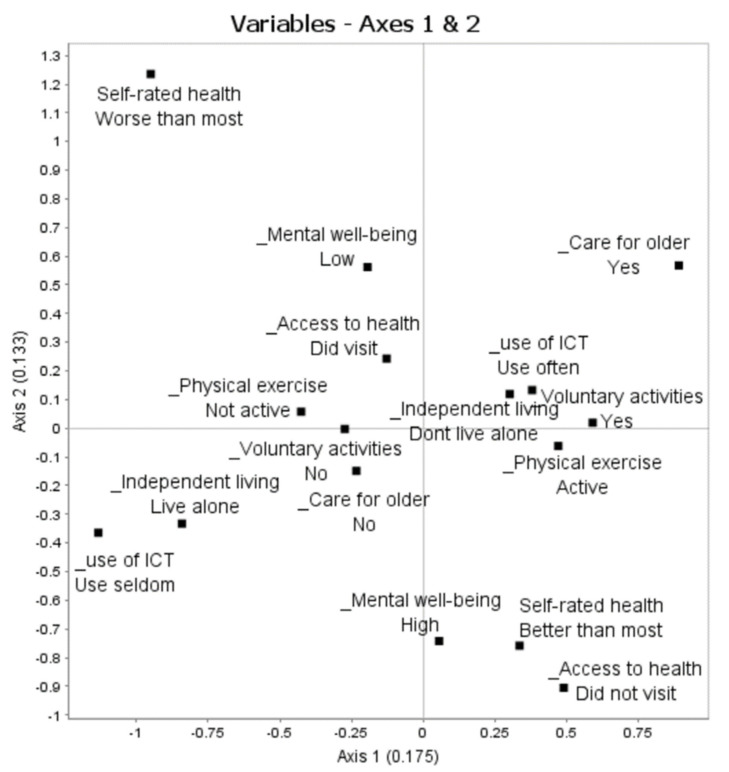
Space of active ageing indicators, Axes 1 and 2. Only response categories with above-average contribution values to one or both axes.

**Figure 2 ijerph-18-09156-f002:**
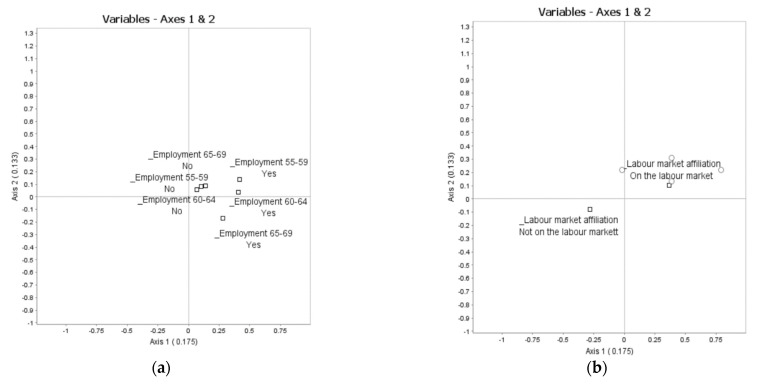
Supplementary variables in the space of active ageing factors. (**a**) Employment by age groups; (**b**) Labour market affiliation; Political participation; (**c**) Political participation; (**d**) Material deprivation; (**e**) Educational attainment; (**f**) Type of employment; (**g**) Financial situation; (**h**) Gender and age.

**Table 1 ijerph-18-09156-t001:** Indicators used to measure inequality in active ageing.

AAI	DSDB	Status in Model	Number of Response Catego-Ries (Active)
*Employment domain*			
Employment rate		Supplementary	
55–59	Employment 55–59: Yes/no		
60–64	Employment 60–64: Yes/no		
65–69	Employment 65–69: Yes/no		
70–74	Employment 70–74: Yes/no		
Labour market affiliation	Are you in the labour market now? Yes/no	Supplementary	
*Participation in society domain*			
Voluntary activities	Do you do volunteer work related to: cultural activities? sports? recreational activities? education? health-related work? social work? the environment? housing and the local community? Yes/no	Active	2
Care for children, grandchildren	Within the last month, have you helped your child/any of your children look after your grandchildren? Yes/no	Active	2
Care for older adults	Within the last month, have you helped your parents/in-laws with cleaning? household chores? finances? health? recreational activities? Yes/no	Active	2
Political participation	Do you do volunteer work? In a trade union, professional association, or business and industry organization? Political work? Yes/no	Supplementary	
*Independent, healthy, and secure living domain*			
Physical exercise	I am now going to mention some ways you can spend your spare time, and I would like to ask you to answer how often you usually do the following: do gymnastics, play sports, exercise, dance? Once or several times/week. Active/not active	Active	2
Access to health care	Have you been to a medical examination or talked to a doctor about your health within the last year? Did visit/did not visit	Active	2
Independent living arrangement:	Do you live alone? Live alone/don’t live alone	Active	2
Relative median income	How do you think your finances are today? Very good/good—OK—bad/very bad	Supplementary	
No severe material deprivation for older persons	Can you afford to invite family and friends to visit once a month? Yes/no Can you afford to participate in the leisure activities you are interested in? Yes/No	Supplementary	
*Capacity and enabling environment for active and healthy ageing domain*			
Remaining life expectancy	Self-rated health: How do you think your health compares to others your age? Worse than most/as good as most/better than most	Active	3
Mental wellbeing	Now, I want to ask about your wellbeing in general. Does it happen often, sometimes, rarely, or never: That you feel well? That you are afraid of certain things? That you are worried? That you are depressed? That you feel lonely? Low/medium/high mental health	Active	3
Use of ICT	I will now mention some ways you can spend your spare time, and I would like to ask you to answer how often you usually do the following: How often do you use the Internet? Daily, once or several times/week, one or more times/month, rarely, never, don’t have internet. Use often/Use rarely	Active	2
Type of employment	What position have you held most of your life? Self-employed, unskilled worker, skilled worker, white collar	Supplementary	
Educational attainment	What vocational training do you have? Basic education, vocational training, short further education (<3 years), longer/further education (3 years or more)	Passive	
*Individual characteristics*			
Gender	Is the interviewee male or female?	Supplementary	
Age	55–59, 60–64, 65–69, 70–74	Supplementary	

**Table 2 ijerph-18-09156-t002:** Interpretation of Axis 1; 8 response categories contributing above-average to axis variance, 4 additional categories contributing below-average added in brackets. Ctr. of variables and categories are in percent.

Axis 1					
Variables	Ctr. of Variables %	Orientation of Categories		Ctr. of Categories %	
		Right	Left	Right	Left
Use of ICT	24.2	Use often	Use seldom	6.7	17.4
Independent living	16.1	Don’t live alone	Live alone	(4.2)	11.9
Care for older	13.2	Yes	No	10.5	(2.7)
Physical exercise	12.7	Active	Not active	6.7	6.0
Voluntary activities	10.3	Yes	No	7.2	(3.2)
Self-rated health	9.8	Better than most	Worse than most	(2.7)	7.1

**Table 3 ijerph-18-09156-t003:** Interpretation of Axis 2; 6 response categories contributing above-average to axis variance, 2 additional categories contributing below-average added in brackets. Ctr. of variables and categories are in percent.

Axis 2					
Variables	Ctr. of Variables %	Orientation of Categories		Ctr. of Categories %	
		Top	Bottom	Left	Right
Self-rated health	37.1	Worse than most	Better than most	15.7	18
Mental wellbeing	23.7	Low	High	11.5	11.0
Access to health care	18.1	Did visit	Did not visit	(3.8)	14.3
Care for older	7.0	Yes	No	5.6	(1.5)

## Data Availability

The raw data used to support the findings of this study are available from the corresponding author upon request.

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
