# Peer review of "Theoretical Challenges and Social Inequalities in Active Ageing"

_ijerph, 2021, doi:10.3390/ijerph18179156_

Round 1

Reviewer 1 Report

The heading and topic of this paper are highly attractive.  Conceptual analyses of the concept of active ageing are a useful addition to the literature, especially when accompanied by data analyses to test the arguments put.

However, I found this article very difficult to review.  One reason may be that the authors’ first language is not English, and therefore it is possible their use of some words and terms does not exactly reflect what they mean.  For example, my dictionary does not include the word “discursivize” and I don’t know which meaning of the word “discursive” it is intended to reflect (i.e., rambling or proceeding by reasoning or argument). Another example is the argument that “active ageing” is primarily a political concept. But the word “political” in English can mean either related to politics or related to policies—two very different things. Finally, it is not at all clear when the authors refer to “construction” whether they mean “construct”.

My first recommendation is that the authors consult a native English speaker about their use of terms in order to ensure that their text says what they mean it to say.

Another, and more important, difficulty is that the authors are clearly setting out with an agenda – to critique the active ageing concept.  Most of their arguments are persuasive.  However, their language is often polemical. The use of phrases like “active ageing machine” and “totalitarian logic” are unnecessarily pejorative and signal bias. 

My second recommendation is that the authors revisit their language and moderate it where appropriate.

Third, I am not convinced by some of the authors' arguments.  This is not to say that the arguments should not remain, only that they may need to be better justified.  For example, the authors claim that the AAU concept will lead to people being forced to continue working.  I can’t see logic of this. There is no external force pressurizing countries to make their older workers remain in the workforce. Currently in Europe a person can’t be prevented from retiring fully or partially.

While the concepts in the AAI are driven from the top down and are necessarily arbitrary, I can’t see that this necessarily disconnects the AAI from “basic notions of society” or “dispositions of older adults”.  The extent to which older adults support the content and weightings of the AAI is an empirical question, but I would think the importance of having sufficient financial security, meaningful things to do, meaningful social engagements, and good mental and physical health would be widely supported.

Nor am I convinced by the argument that the assumptions that underlie the Index are faulty because it isn’t possible to reach 100% (line 139).  This is too blunt an argument. The optimal level of activity as a predictor of quality of life for individuals in different age groups is an empirical question. Theoretically one might expect that an inverted u-shape might best describe the relationship between activity and wellbeing.

My third recommendation is that the authors ensure that their arguments are supported better and/or are more nuanced.

The authors say that they set out to test empirically whether the AAI active ageing concept is realistic and for whom, but I could not see any conclusion on this question arising from the statistical analyses, or even how the analyses addressed this question.  The authors say in their conclusion that the normative judgements underlying the AAI are idealistic rather than realistic.  This is likely true, but it’s not clear from the data analysis how they come to this conclusion.  Further, the authors say that the AAI ignores the life situation of large segments of the population in that active ageing opportunities are conditional on one’s position in the social structure.  Again, this is indisputably true, but surely this is the whole point of doing country-wide comparisons of the AAI – to encourage countries to develop policies that facilitate participation, by addressing social disadvantage and removing barriers to participation.

My fourth recommendation is related to the third -- that the authors revisit both the strengths and weaknesses of the AAI approach and ensure that their conclusions are evidence based.

The analyses are difficult to understand. The three axes derived from the research are meant to be orthogonal in three-dimensional space, yet they clearly overlap conceptually and on the measures that load on them.  The axes and the labels given to them are difficult to distinguish conceptually. How is General health and active living (characterized by the high use of ICT) different from Health (Axis 2 – distinguished from Axis 1 by not having any caregiving responsibilities for an older person) and Physical and social activity (Axis 3)?  Would the axes be more meaningful if they weren’t constrained to be orthogonal?  Can better labels be identified to distinguish the axes?  Would the axes be better described as an overall “factor” and two “sub-factors” (i.e., 1 = overall health and activity, with 2 = health and 3 = activity as subcomponents)?

The figures are difficult to read and interpret and some labels appear at some distance from the dot point they are attached to (at least I think this is the case – see the label Use of ICT Use often).

My fifth recommendation is that the authors revisit their descriptions of the results of their analyses and their presentation to see if they can make them more comprehensible.

The discussion does not admit to any weaknesses in the research, whereas there are several.  One of the most serious in my view is the measure of living independently, which in the AAI is living alone or in a couple household, not just living alone. (It might also make better sense if the AAI looked at proportion of older people not living in institutional care, but that’s another argument.)

My sixth recommendation is that the authors include a section in their discussion that describes the strengths and weaknesses of their research.

Before I go on to the seventh recommendation, I would like to comment on a feature of the conceptualization of active ageing that is missing (or perhaps disguised) in the article – and that is, the interesting tension between activity as a contribution to the social economy of a nation and activity as a contribution to individual wellbeing.  In the case of caring for a disabled spouse, for example, these contributions pull in different directions. 

Another comment I would make is that the importance of access to IT is really interesting – and has become increasingly so under COVID restrictions.

My seventh recommendation (see below) is that the authors obtain the assistance of a native English speaker on the wording of their article and proofread it carefully before resubmission.

The version I read (in pdf) might have removed some formatting and spaces, so I won’t comment on these flaws. 

Please spell ageing consistently throughout (either ageing or aging but not both unless in direct quotes).

A whole sentence is repeated after line 305.

Please pay attention to the following examples, all derived from just the first two pages of the text.  The whole text needs a thorough edit but it would take too long to do this here.

Abstract

Suggested rewording: . . . showing how the conceptual framework of active ageing is undertheorized, lacks conceptual and analytical clarity, and fails to propose clear contributing factors and barriers.

Introduction

Write: by rejecting the view that old age

Consider: Barrio et al. claimed that the discourse of active ageing has paved the way for active citizenship. In contrast, other research has highlighted lack of alignment between social policies encouraging active ageing and older adults’ wishes and aspirations regarding retirement. (Note the apostrophe.)

The sentence beginning on line 44 makes no sense. (Perhaps you meant “a” rather than “at”.)

Write: The AAI project occupies

Write: The aims of this paper are, first, to make a critical assessment of how the AAI project has conceptualized active ageing and, second, to analyze social inequalities in active ageing empirically.

Write: It has been proposed that norms of active ageing . . . and that social inequalities condition opportunities for active ageing.  Following these hypotheses . . .

Write: given that the active ageing concept comprises four domains

Write: to explore interdependency between the domains empirically.

Rewrite the sentence on lines 86-88. (The phrase “social inequalities conditional opportunities” is unreadable.)

Write: comprising 9,965 interviews

Consider: data are plural, hence, “the data used in this paper are an approximation”

Write: active ageing should neither be studied independently nor be reduced to a single dimension

Rewrite the sentence on lines 97-100.  What does “will be indicative of a call for a social theorizing of (inequalities in) active ageing” mean?  Perhaps, “will provide evidence to underpin a theorization of”

Write line 103 limitations (i.e., plural -- and ensure you do so!)

Reviewer 2 Report

The authors addressed a very interesting topic and they proposed a very innovative way to discuss Active ageing. I suggest to better describe, in the first part of the article, why they choose this approach and, in the conclusion, what are the consequences of this approach. 

Some minor changes are needed in references that are quoted both by numbers and by roman numbers. I do not understand why. May there are some notes? 

Reviewer 3 Report

Theoretical Challenges and Social Inequalities in Active Ageing

Per H Jensen and Jakob Skjøtt-Larsen

I encourage this type of work that promotes theory based on empirical studies. Theory is important in the progress of scientific studies.  This study by the authors fulfills this interest. The primary fault in the paper is the emphasis and secondary and peripheral but still important is the use of English that detract from the important findings of this paper. I will attempt to provide criticisms that will hopefully address these shortfalls and make suggestions for editing.

The emphasis of the paper is wrong. I understand why the authors first wanted to dispel the idea of a metric that measures an ideal state of aging. I share their sentiments, but this study just comes up with a different metric rather than dispelling the idea of a metric. The confusion starts with the cited objective of the paper. There are so many of them

LINES 58-60

The aim of this paper is, first, to make a critical assessment as to how the AAI project 58 has conceptualized the construction of active ageing. Second, to analyze empirically social 59 inequalities in active ageing.

LINES 64-65

Our main argument is that the AAI-active ageing construction is initiated by international political organizations, i.e. it is a political project rather than a scientific one.

LINES 73-78

Following this hypothesis a central aim of this paper is empirically, first, to test whether AAI-active ageing is realistic – and for whom. Furthermore, given that the AAI-active ageing concept is composed by different domains of active ageing, i.e. 1) employment, 2) participation in society, 3) independent living, and 4) capability to age actively, a second aim is to explore empirically interdependency between different domains.

LINES 78-79

Not least, we explore potential Matthew Effects of accumulated advantage

LINES 86-88

…we analyze the multidimensionality in active ageing and identify potential structuring factors, bringing social inequalities conditioning opportunities for active ageing to the forefront of the analysis.

I understand the aim of this paper to be: Assessing how the AAI project has conceptualized the construct of active ageing and using empirical data to define a more realistic definition of the construct.

It would be helpful to the readers if the paper is focused on a simple objective. There are many criticisms of the AAI project, and you rightly need to put these criticisms in the paper, but in a different format rather than as the objective of this paper. Keep the objective of this paper simple and straightforward that reflect the actual work done.

I understand why you criticize the AAI project

LINES 95-97

The methodological approach underlines the theoretical argument, that active ageing should neither be studied independently nor should it a priori be reduced to a single dimension.

And

LINE 223 AAI project is that it neglects differences in value systems and cultural orientation across Europe

However even though again you are right, you come up with your own metric and with your own (different) culture. Admit this limitation. The AAI project was designed to develop a metric, a scale. Like any metric, then as you report (LINE 127) “more is better” value judgment.  This is what a metric does, it measures a construct on a scale and more is better. In your conclusion, you came up with your own metric composed of three components. Although you did not combine them or weight them, they are independent metrics of a construct “active ageing.” What you developed is a metric, whether you like it or not.

The AAI project has four components 1) employment, 2) participation in society, 3) independent living, and 4) capability. LINE 194-198 For instance, they have found a strong correlation linking Domain 4 (capability to age actively) to Domain 2 (participation in society) and Domain 3  (independent living) [13]. This points in the direction that different activities are interdependent [18], meaning that a high score in one domain is associated with high scores in other domains.

Your study came up with three components 1) general health and active living, 2) mental and physical health, and 3) physical and social activity. There are some similarities but there is one distinct difference. You argue that your three factors (Axis?) are orthogonal projections and are uncorrelated. This is an important finding and needs to be better highlighted. That the empirically derived components/factors/axis are independent and therefore they are better able to differentiate the population. Perhaps with LINES 467-469

…this study has found a clear Matthew Effect of accumulated advantage: Those scoring high on one indicator/domain also score high on other indicators/domains.

The AAI project develops an aggregate metric which you then criticize on an individual level. Your primary complaint can be summarized under three points. Individuals might be unable to perform these active ageing components, it might not be possible for them to engage in all four dimensions at the same time, and they might be unwilling to perform any or all of them. These are valid criticisms, but would they also be valid for your own study? Address this in your conclusion.

General criticism of your study.

Because of the database you use, the select group is skewed, in your own estimation, to be educated and having no mental health issues. Now that you developed your own components/factors/axis it would have been theoretically more useful to look at a group of disabled and older adults with mental health issues to review whether the active aging components/factors/axis are valid. The choice of the database you selected, therefore, limits the theoretical horizon. As the authors themselves acknowledge in LINE 458 when criticizing the AAI project that also applies to their study:

The active ageing approach tends to neglect frailty and limitations

The paper instead of trying to contradict active aging should have argued for a modification, an improvement upon the existing structure to allow for the greater empirical contribution that will promote more research as you say.

LINE 451

rooted in evidence-based, empirical research

Some proof editing:

  1. Citation numbers are confusing and inconsistent eg on LINE 163 [13, xxxiii] LINE 292 [xliv] my suggestion is to change them all to Roman numerals.
  2. LINE 8 discursivized no idea what this word means. Divergent, discordant?
  3. LINE 268 whith
  4. Define ICT as Internet Use.

In summary, the edits that would help this paper:

  1. Summarize and simplify the objectives of the paper
  2. Criticize the AAI project and then introduce how you are going to improve on the metric
  3. Highlight why this empirical approach is better and admit your limitations
